# The Reducing Agent Dithiothreitol Modulates the Ventilatory Responses That Occur in Freely Moving Rats during and following a Hypoxic–Hypercapnic Challenge

**DOI:** 10.3390/antiox13040498

**Published:** 2024-04-22

**Authors:** Paulina M. Getsy, Gregory A. Coffee, Walter J. May, Santhosh M. Baby, James N. Bates, Stephen J. Lewis

**Affiliations:** 1Department of Pediatrics, Case Western Reserve University, Cleveland, OH 44106, USA; pxg55@case.edu (P.M.G.); gac43@case.edu (G.A.C.); 2Department of Pediatrics, University of Virginia, Charlottesville, VA 22903, USA; walterjmay@gmail.com; 3Galleon Pharmaceuticals, Inc., 213 Witmer Road, Horsham, PA 19044, USA; santhosh.m.baby@galvani.bio; 4Department of Anesthesiology, University of Iowa Hospitals and Clinics, Iowa, IA 52242, USA; jbates@atelerixlifesciences.com; 5Department of Pharmacology, Case Western Reserve University, Cleveland, OH 44106, USA; 6Functional Electrical Stimulation Center, Case Western Reserve University, Cleveland, OH 44106, USA

**Keywords:** oxidation–reduction, dithiothreitol, N-acetyl-L-cysteine methyl ester, hypoxic–hypercapnic challenge, ventilation, rats

## Abstract

The present study examined the hypothesis that changes in the oxidation–reduction state of thiol residues in functional proteins play a major role in the expression of the ventilatory responses in conscious rats that occur during a hypoxic–hypercapnic (HH) gas challenge and upon return to room air. A HH gas challenge in vehicle-treated rats elicited robust and sustained increases in minute volume (via increases in frequency of breathing and tidal volume), peak inspiratory and expiratory flows, and inspiratory and expiratory drives while minimally affecting the non-eupneic breathing index (NEBI). The HH-induced increases in these parameters, except for frequency of breathing, were substantially diminished in rats pre-treated with the potent and lipophilic disulfide-reducing agent, L,D-dithiothreitol (100 µmol/kg, IV). The ventilatory responses that occurred upon return to room air were also substantially different in dithiothreitol-treated rats. In contrast, pre-treatment with a substantially higher dose (500 µmol/kg, IV) of the lipophilic congener of the monosulfide, N-acetyl-L-cysteine methyl ester (L-NACme), only minimally affected the expression of the above-mentioned ventilatory responses that occurred during the HH gas challenge or upon return to room air. The effectiveness of dithiothreitol suggests that the oxidation of thiol residues occurs during exposure to a HH gas challenge and that this process plays an essential role in allowing for the expression of the post-HH excitatory phase in breathing. However, this interpretation is contradicted by the lack of effects of L-NACme. This apparent conundrum may be explained by the disulfide structure affording unique functional properties to dithiothreitol in comparison to monosulfides. More specifically, the disulfide structure may give dithiothreitol the ability to alter the conformational state of functional proteins while transferring electrons. It is also possible that dithiothreitol is simply a more efficient reducing agent following systemic injection, although one interpretation of the data is that the effects of dithiothreitol are not due to its reducing ability.

## 1. Introduction

Protocols involving the exposure to hypoxic (HX), hypercapnic (HC), or hypoxic–hypercapnic (HH) gas challenges and the subsequent return to room air are employed to study the central and peripheral components of ventilatory control systems in health and disease [1,2,3,4,5,6,7,8,9,10,11,12], and to evaluate how drugs, such as opioids, affect ventilatory systems [13,14,15,16,17,18,19,20]. Short-term exposures to HX [11,14,16,21,22,23,24,25,26,27,28,29], HC [16,21,22,23,30,31], or HH [10,11,16] gas challenges elicit robust changes in ventilatory parameters in rats and mice. In addition, excitatory and inhibitory ventilatory responses occur upon return to room air following HX, HC, or HH gas challenges [10,15,16,24,25,26,27,28,29,30,31]. Ventilatory responses that occur during HX gas challenges and upon the return to room air are markedly diminished in freely moving rats and mice with bilateral carotid sinus nerve transection (CSNX), demonstrating the vital role of the carotid body–CSN complex in the expression of these responses [11,26,30]. Although HC gas challenges activate the carotid body–CSNX complex, the maximal ventilatory responses that occur during HC gas challenges are minimally impaired by CSNX [26,30] because of the ability of CO_2_ to initiate centrally mediated ventilatory responses [31,32,33]. However, the ventilatory responses that occur upon return to room air following HC gas challenges are markedly reduced by CSNX, thus demonstrating that the carotid body–CSN complex plays a vital role in the expression of these return-to-room-air responses [26,30]. 

Short-term exposure to HX [34,35,36,37,38,39], HC [40,41,42,43], or HH [44,45] gas challenges have dramatic effects on the redox (oxidation–reduction) status of the brain and other tissues. Although numerous studies have demonstrated profound redox changes in the carotid bodies [46,47,48,49,50], little is known about potential redox changes that occur in response to acidosis or hypercapnia [37]. Moreover, there are ongoing disputes about the exact nature of redox changes that occur during HX gas challenges and the exact roles of reactive oxygen species (ROS) and the electron transport chain [49,50,51,52,53,54,55,56,57]. Similarly, although reoxygenation elicits a burst of ROS in tissues, such as the carotid bodies, the physiological importance of ROS in the functioning of these cells remains in dispute [35]. Moreover, there is now substantial evidence that a general shift in intracellular redox potential caused by HX, HC, and HH challenges is not the primary stimulus that leads to the depolarization of carotid body glomus cells and subsequent neurotransmitter release from these cells [51,52,53,54,55].

Unfortunately, little is known about the redox changes that occur in the carotid bodies upon return to room air after HX, HC, or HH gas challenges in vivo. However, whereas Bernardini et al. [50] found that the intracellular redox potential of isolated mouse carotid body primary glomus cells became more oxidized upon reoxygenation after HX challenges, which in itself drove a minor reductive change in redox status, Roy et al. [58,59] reported that the application of the powerful dithiol-reducing agent, L,D-dithiothreitol (L,D-DTT) [60,61,62,63]-the reduced form of L-glutathione, or cytochrome P-450 inhibitors did not influence HX-induced chemosensory responses in an isolated rat carotid body–carotid sinus nerve preparation. Another factor to be considered is that HX, HC, or HH gas challenges cause a change in the redox status of functional proteins that are sustained during the reoxygenation period. For example, Tawa et al. [64] reported that the ability of nitric oxide to activate soluble guanylate cyclase (SGC) is impaired during HX and especially upon reoxygenation by mechanisms involving the generation of intracellular superoxide anions by sources that were different during and after HX gas challenge. Moreover, Tawa et al. [65] reported that the nitric oxide-sensitive reduced form of SGC converts to a nitric oxide-insensitive oxidized/heme-free form under oxidative stress and that the oxidized status of SCG was fully maintained upon reoxygenation.

We are interested in determining whether the redox status of both peripheral and central ventilatory control structures (e.g., carotid body and brainstem nuclei) plays a role in the expression of the ventilatory responses to HX, HC, and HH gas challenges and the return-to-room-air phase after these challenges, which is associated with a pronounced degree of non-eupneic breathing (NEBI) including, apneas and episodes of disordered breathing [10,27,28]. As such, developing redox-modulating drugs that can reduce NEBI may have therapeutic value for the treatment of conditions associated with irregular breathing patterns, such as sleep apnea [66,67], and in subjects using opioids, which are known to greatly increase NEBI [19,20]. The main aim of this study was to see whether the intravenous administration of L,D-DTT to unanesthetized rats would influence the ventilatory responses that occur during a HH gas challenge and upon return to room-air. L,D-DTT, which can interchange between reduced (free sulfurs) and oxidized (disulfide bond) forms (Figure 1), has been given to experimental animals to address the role of redox status in a variety of conditions and disease states [68,69,70,71,72,73,74,75]. To build on information gathered from the L,D-DTT study, we examined whether pretreatment with N-acetyl-L-cysteine methyl ester (L-NACme, Figure 1), which is a cell-penetrant L-thiolester reducing agent [76,77,78,79,80,81,82,83,84,85], was able to mimic the effects of L,D-DTT. The present findings add to our knowledge about the pharmacological actions of L,D-DTT and L-NACme by showing that L,D-DTT, but not L-NACme, had major effects on the expression of many of the ventilatory responses that occurred during and following a HH gas challenge.

## 2. Materials and Methods

### 2.1. Permissions, Rats, and Surgical Procedures

The experiments described in this manuscript were performed according to the NIH Guide for Care and Use of Laboratory Animals (NIH Publication No. 80-23) revised in 1996 and to ARRIVE (Animal Research: Reporting of In Vivo Experiments) guidelines. All experimental protocols were approved by the Animal Care and Use Committees of the University of Virginia (3642-09-07), Galleon Pharmaceuticals (PC0022), and Case Western Reserve University (2015-0005). Adult male Sprague Dawley rats with indwelling femoral vein catheters were purchased from Harlan Industries (Madison, WI, USA). The rats were given 4 days to recover from transport before use. On the day of study, the venous catheter was flushed with a heparin solution (50 units heparin) in 0.1 M, pH 7.4 phosphate-buffered saline 3–4 h before commencing the study. Stock solutions of the vehicle or L,D-DTT were brought to pH 7.2 with 0.25 M NaOH. All studies were conducted in a quiet room with a temperature of 21.2 ± 0.2 °C and relative humidity of 51 ± 2%.

### 2.2. Protocols for Whole-Body Plethysmography Measurement of Ventilatory Parameters

Whole-body plethysmography (PLY3223; Data Sciences International, St. Paul, MN, USA) was used to record ventilatory parameters in freely-moving rats [10,11,14,15,16,19,20,27,28,29,30,31]. Directly recorded and calculated/derived parameters are defined in Table 1 [85,86,87,88,89]. Ventilatory parameters with their abbreviations are as follows: frequency of breathing (Freq), tidal volume (TV), minute ventilation (MV), inspiratory time (Ti), expiratory time (Te), Ti/Te, peak inspiratory flow (PIF), peak expiratory flow (PEF), PIF/PEF, inspiratory drive (TV/Ti, InspD), expiratory drive (TV/Te, ExpD), non-eupneic breathing index (NEBI), and NEBI corrected for Freq (NEBI/Freq). Just prior to starting the protocols, each rat was placed in a plethysmography chamber and given 60 min to become familiar with the chambers to allow baseline values to be accurately assessed. The rats explored the chambers for 5–10 min before settling in a resting position up against the chamber wall. The HH gas challenge consisted of stopping the 1.5 mL/min flow of room air through the chambers for 60 min [16]. This allowed the rats to gradually rebreathe their own air which became progressively lower in O_2_ and higher in CO_2_, thereby representing a physiologically relevant HH gas challenge. There were minimal overt changes in behavior during this time, although there was a general increase in movement about the chamber, rearing and other signs of arousal at 55–60 min. After 60 min, the flow of room air was restored, and ventilatory parameters were recorded for another 30 min. There were minor changes in body temperature (Table 2), and in-built recording systems and FinePointe (DSI) software (v2.9.0) continuously corrected flow parameters (e.g., TV, PIF, PEF) for changes in chamber humidity and chamber temperature [15,16]. **Study 1:** One group of rats (n = 6, 326 ± 2 g) received an injection of vehicle (100 μL/kg, IV) and another (n = 6, 327 ± 2 g) received an injection of L,D-DTT (100 μmol/kg, IV). After 15 min, both groups received a HH gas challenge for 60 min before restoring the flow of room air. **Study 2:** One group of rats (n = 6, 324 ± 3 g) received an injection of vehicle (100 μL/kg, IV) and another (n = 6, 323 ± 3 g) received an injection of L-NACme (500 μmol/kg, IV). After 15 min, both groups were subjected to a 60-min HH gas challenge before restoring the flow of room-air. Note that two separate baches of 3 rats per group were used in each of the above studies to help ensure that the data was reproducible. As the body weights of all groups of rats were similar to one another (see above), ventilatory parameters related to volumes (e.g., TV, PIF, PEF) are presented without correcting for body weights. The incidence of non-eupneic breathing events, such as apneas, type 1 sighs and type 2 sighs, and irregular breaths (% of each 1 min recording epoch) are expressed as the non-eupneic breathing index (NEBI) and NEBI/Freq (NEBI corrected for the level of Freq) [10].

### 2.3. Body Temperatures

We recorded rat body temperatures before (Pre-drug) and at various timepoints (Post-drug) following the injection of vehicle (saline; n = 6 rats, 328 ± 3 g) or L,D-DTT (100 µmol/kg, IV; n = 6 rats, 327 ± 7 g). We also recorded rat body temperature at various timepoints during a 60-min HH gaschallenge, and upon return to room-air (Post-HH gas challenge) via a rectal thermometer (Data Sciences International, St. Paul, MN, USA) as detailed previously [19,85,90]. In brief, a thermistor probe was inserted 5–6 cm into the rectum to allow for regular recordings of body temperature. A 2–3 inch length of the probe cable was taped to the tail. The rats were then placed in the plethysmography chambers, and the probe cable was exteriorized to the chambers via a side port on the chambers and connected to a telethermometer (YellowSprings Instruments, South Burlington, VT, USA). The rats were allowed 45–60 min to acclimatize.

### 2.4. Data Analyses

All the data collected in these studies are presented as mean ± SEM. The data from each set of experiments were statistical analyzed by one-way or two-way ANOVA followed by Bonferroni corrections for multiple comparisons between means using the error mean square terms from each individual ANOVA analysis [91,92,93,94] as described in detail previously [27,28,29,30]. A *p* < 0.05 value provided the initial level of statistical significance that was modified according to the number of comparisons between means as described in detail by Wallenstein et al. [91]. The statistical analyses were carried out using GraphPad Prism software version 9 (GraphPad Software, Inc., La Jolla, CA, USA).

## 3. Results

### 3.1. Ventilatory Baseline Values and Effects of L,D-DTT

Baseline (Pre) ventilatory values for the two groups of rats, which would eventually receive a bolus injection of vehicle or L,D-DTT (100 µmol/kg, IV), are summarized in Table 3 and Table 4. There were no between-group differences for any parameter. Table 3 and Table 4 also summarize (a) the values for the ventilatory parameters at time of maximum change elicited by an injection of vehicle or L,D-DTT (designated in the column as Drug Max), (b) the values immediately before starting the HH gas challenge (Pre-HH), (c) the maximal values reached during the HH gas challenge (HH Max), (d) the values immediately before re-starting the flow of air in the plethysmography chambers (Pre-RA), and (d) the maximal responses that occurred upon the return-to-room-air challenge (RA Max). Key details pertaining to the data in Table 3 and Table 4 will be addressed in the following descriptions of the changes in the individual ventilatory parameters.

### 3.2. Effects of L,D-DTT on Changes in Frequency of Breathing, Tidal Volume, and Minute Ventilation

Freq, TV, and MV values before (Pre), during the HH gas challenge, and upon return to room air (Post) in vehicle- or L,D-DTT (100 µmol/kg, IV)-treated rats are summarized in Figure 2. The injection of L,D-DTT elicited a substantial but relatively brief increase in, but not TV, which translated into a robust increase in MV. The HH gas challenge in the vehicle-injected rats elicited increases in Freq (Figure 2A, TV (Figure 2B), and MV (Figure 2C). The pattern of increases in Freq were similar in both the vehicle- or L,D-DTT-treated rats, whereas the increases in TV and MV were smaller in the L,D-DTT-treated rats. In the vehicle-treated rats, return to room air resulted in a transient and relatively sustained increase in Freq, and a transient drop towards baseline and then slight increase in TV which translated into a pronounced and sustained elevation in MV that took about 15 min to subside. It is important to note that the responses were not associated with overt changes in the behavior of the rats [16]. These changes in return-to-room-air responses were absent in the L,D-DTT-treated rats for Freq, TV and MV as the values returned rapidly to baseline within 2–3 min.

### 3.3. Effects of L,D-DTT on Changes in Inspiratory Time and Expiratory Time

Values for Ti, Te, and Ti/Te before (Pre), during the HH gas challenge, and upon return to room air (Post) in vehicle- or L,D-DTT (100 µmol/kg, IV)-treated rats are shown in Figure 3. The injection of L,D-DTT elicited brief falls in Ti (Panel Figure 3A) and very pronounced falls in Te (Panel Figure 3B) that translated into substantial, but brief increases in Ti/Te (Panel Figure 3C). The HH gas challenge in the vehicle-injected rats elicited sustained decreases in Ti and more substantial decreases in Te, resulting in marked increases in Ti/Te. HH-induced responses were similar in the L,D-DTT-injected rats compared to the vehicle-injected rats. In the vehicle-treated rats, return to room air resulted in a gradual recovery of Ti and Te to at or above baseline values with rapid normalization of Ti/Te to pre-HH values. The recovery of Ti and, to a lesser extent, Te occurred more rapidly in the L,D-DTT-treated rats compared to the vehicle-treated rats, although the return to pre-HH Ti/Te values occurred quickly in the DTT group as in the vehicle group.

### 3.4. Effects of L,D-DTT on Changes in Peak Inspiratory Flow and Peak Expiratory Flow

The values for PIF, PEF, and PIF/PEF before (Pre), during the HH gas challenge, and upon return to room air (Post) in vehicle- or L,D-DTT (100 µmol/kg, IV)-treated rats are summarized in Figure 4. The injection of L,D-DTT elicited similar changes in PIF, PEF, or PIF/PEF compared to the injection of vehicle. The HH gas challenge in the vehicle-injected rats elicited robust increases in PIF (Figure 4A) and in PEF (Figure 4B) such that there were sustained decreases in PIF/PEF (Figure 4C). The increases in PIF and PEF were much smaller in the L,D-DTT-treated rats compared to vehicle-treated rats, whereas the fall in PIF/PEF was similar to that in the vehicle-treated rats. In the vehicle-treated rats, the return to room air resulted in a gradual return of PIF to near baseline values, and a more prompt recovery of PEF to baseline values, which therefore translated into a fairly rapid recovery of PIF/PEF values to near baseline levels. Return-to-room-air PIF and PEF values recovered more quickly in the L,D-DTT-injected rats compared to the vehicle-injected rats, however the recovery of PIF/PEF values occurred in a similar fashion compared to the vehicle-injected rats.

### 3.5. Effects of L,D-DTT on Changes Inspiratory Drive and Expiratory Drive

Inspiratory drive and expiratory drive values before (Pre), during the HH gas challenge, and upon return to room air (Post) in vehicle- or L,D-DTT (100 µmol/kg, IV)-treated rats are summarized in Figure 5. The injection of L,D-DTT elicited minor increases in inspiratory and expiratory drives. The HH gas challenge elicited robust increases in inspiratory drive (Figure 5A) and expiratory drive (Figure 5B) in the vehicle-injected rats. The increases were smaller in the L,D-DTT-treated rats. In the vehicle-treated rats, return to room air resulted in a gradual return of inspiratory drive and a fast recovery of expiratory drive to near baseline values. In contrast, the return to baseline values happened more rapidly in the L,D-DTT-treated rats.

### 3.6. Effects of L,D-DTT on Changes in Non-Eupneic Breathing Index

Values for NEBI and NEBI/Freq before (Pre), during the HH gas challenge and upon return to room air (Post) in vehicle- or L,D-DTT (100 µmol/kg, IV)-treated rats are summarized in Figure 6. The injection of vehicle or L,D-DTT elicited minor decreases in NEBI (Figure 6A) and NEBI/Freq (Figure 6B), noting that levels of non-eupneic breathing were minor in both groups. The HH gas challenge elicited negligible responses in either group. Upon return to room air, the vehicle-treated rats displayed rapid dramatic increases in NEBI and NEBI/Freq of about 10–15 min in duration before quickly returning to near baseline values. These return to room-air responses were absent in the L,D-DTT-treated rats.

### 3.7. Summary of the Effects of L,D-DTT

The data in Table 5 confirms that the time taken to recover to baseline upon return to room air following the HH gas challenge was markedly different in L,D-DTT- compared to vehicle-injected rats. A summary of the total (cumulative) responses that occurred during the first 30 min (Figure 7A) and the second 30 min (Panel B) period of the total 60 min HH gas challenge in vehicle- or L,D-DTT (100 µmol/kg, IV)-treated rats are summarized in Figure 7. Over the first 30 min of the HH gas challenge, the increases in MV, PIF, inspiratory drive, and expiratory drive were smaller in the L,D-DTT-treated rats compared to the vehicle-treated rats, whereas all other ventilatory responses were similar in both groups. Over the second 30 min of the HH gas challenge, the increases in MV, PIF, PEF, inspiratory drive, and expiratory drive were smaller in the L,D-DTT-treated rats compared to the vehicle-treated rats, whereas all other ventilatory responses were similar in both groups. A summary of the cumulative responses that occurred over the entire 60-min HH gas challenge is shown in Figure 8. The increases in MV, PIF, PEF, and inspiratory and expiratory drives were smaller in the L,D-DTT-treated rats compared to the vehicle-treated rats, whereas all other ventilatory responses were similar in the two groups. A summary of the total responses that occurred during the 30 min period following return to room air after the 60 min HH gas challenge in rats injected with vehicle (VEH) or L,D-DTT (DTT, 100 µmol/kg, IV) is shown in Figure 9. The total changes in Freq, TV, MV, Ti, PIF, PEF, InspD, ExpD, NEBI, and NEBI/Freq were markedly less in the DTT-treated rats than in the vehicle-treated rats. The changes in Te, Ti/Te, and PIF/PEF were similar in both groups. The body temperatures recorded before and following the injection of vehicle or L,D-DTT (100 µmol/kg, IV) during and following the 60 min HH gas challenge are summarized in Table 2. The injection of L,D-DTT elicited a minor fall in body temperature, and the HH gas challenge elicited minor hypothermia in the vehicle-injected rats, but did not affect the minor hypothermia already present in the L,D-DTT-treated rats. Body temperatures returned rapidly to Pre-values upon return to room-air in both the vehicle- and L,D-DTT-injected rats.

### 3.8. Effects of L-NACme on Ventilatory Responses

As shown in Figure 10, the injection of L-NACme (500 µmol/kg, IV) elicited a prompt, but relatively transient, increase in Freq (Figure 10A) and MV (Figure 10C), but no change in TV (Figure 10B). The increases in Freq, TV, and MV during the HH gas challenge were similar in the vehicle- and L-NACme-injected rats. The return-to-room-air responses were also similar in both groups. As seen in Figure 11, L-NACme (500 µmol/kg, IV) elicited prompt, relatively transient increases in NEBI (Figure 11A) and NEBI/Freq (Figure 11B). The HH gas challenge minimally affected NEBI or NEBI/Freq in both groups. Upon return to room-air, the vehicle- or L-NACme-treated rats displayed equally robust increases in NEBI and NEBI/Freq for about 15–20 min.

## 4. Discussion

The present study demonstrates that the intravenous injection of L,D-DTT (100 μmol/kg, IV) elicited (1) relatively substantial but brief increases in Freq that were associated with relatively greater falls in Te than Ti such that the respiratory quotient (Ti/Te) rose substantially, (2) only minimal changes in TV, but substantial increases in Freq, resulting in increased MV, and (3) minor changes and PIF, PEF, PIF/PEF, inspiratory and expiratory drives, NEBI, and NEBI/Freq. All the responses to L,D-DTT injection had fully resolved when the HH gas challenge started, so the ability of L,D-DTT to have such profound effects on ventilatory responses that occurred during the HH gas challenge and upon return to room-air is remarkable. The intravenous injection of L-NACme (500 μmol/kg, IV) also elicited a robust transient increase in Freq and MV that was not associated with a change in TV. The lack of effect of L-NACme on the HH gas challenge and return-to-room-air ventilatory responses is also somewhat remarkable considering the robust increases in redox equivalents and concentrations of cysteine and glutathione produced by such injections [76,77,78,79,80,81,82,83,84]. We do not know the mechanisms by which L,D-DTT and L-NACme elevate Freq, but speculate that they may involve effects within the carotid body complex that may be absent in rats with bilateral carotid sinus nerve transection. There are many caveats to the discussion as to why L,D-DTT, but not L-NACme, was effective in modulating the ventilatory responses to HH gas challenge, and those upon return to room-air. First, it is possible that effective concentrations of L,D-DTT may have been sustained throughout the experimental period, whereas those of L-NACme (and downstream products) may not have been maintained at bio-effective levels. Second, it should be noted that the activity of L,D-DTT may not necessarily be related to its ability to alter redox status, but rather to L,D-DTT interacting directly with plasma membrane and/or intracellular signaling proteins. More specifically, Alliegro [61] found that the actions of L,D-DTT on protein activity were unrelated to thiol–disulfide exchange processes. Third, the simple explanation may be that the biological activity of L,D-DTT is related to its ability to become the oxidized form with a disulfide bond (Figure 1) that exerts direct effects on signaling processes, including direct docking to proteins that may also involve redox (electron transfer) processes, resulting in the oxidation of the protein and reduction of oxidized L,D-DTT in a manner that cannot be replicated by the disulfide form of L-NACme. A fourth possibility is that L,D-DTT is nitrosylated in vivo to produce the biologically active cell-penetrant S-nitrosothiol, S-nitroso-L,D-DTT. Brock et al. [95] found that bath-applied S-nitroso-L,D-DTT is a potent inhibitor of voltage-gated K^+^ (Kv) channels and provided evidence that it crossed cell membranes to block Kv channels at an internal site likely in the channel cavity. S-nitroso-L,D-DTT was selective for Kv1 channels since Kv1.1-1.6 displayed profound time-dependent blocks by the S-nitrosothiol, whereas Kv2.1, 3.1b and 4.2 were unaffected.

The present study confirms that the rebreathing method to expose unanesthetized rats to a HH gas challenge elicited gradual and eventually substantial changes in ventilatory parameters that were accompanied by minimal changes in body temperature [16]. Upon the cessation of the continuous airflow to a plethysmography chamber with a rat inside, it is expected that O_2_ concentrations in the chamber progressively drop, and CO_2_ levels progressively rise. These changes in chamber O_2_ and CO_2_ concentrations lead to arterial blood in the rat becoming progressively hypoxemic and hypercapnic. As such, the HH gas challenge is expected to activate carotid body–carotid–sinus–chemoafferents [9,48,58,59] and brainstem structures, such as the retrotrapezoid nucleus–parafacial complex [31,32,33,96]. The pattern of ventilatory responses during the HH gas challenge in the vehicle-injected rats included (1) increases in Freq associated with relatively greater decreases in Te than Ti such that the respiratory quotient (Ti/Te) rose considerably, (2) increases in TV and therefore increases in MV, (3) relatively greater increases in PEF than PIF such that PIF/PEF values fell, (4) increases in inspiratory drive and expiratory drive, and (5) relatively minor changes in NEBI and NEBI/Freq. To what extent the ventilatory responses involve peripheral (e.g., carotid body chemoreceptors) and/or central processes is undetermined. The HH-induced increase in TV was markedly smaller in L,D-DTT-treated rats, whereas the increases in Freq were similar in vehicle- and L,D-DTT-treated rats. The intriguing finding that L,D-DTT differentially affects these parameters suggests that L,D-DTT-sensitive processes take place in central/peripheral systems that drive TV but not in those that drive Freq during HH gas challenge. With respect to carotid body function, it is known that whereas redox changes in functional proteins, such as K^+^ channels, occur in carotid bodies upon HX exposure, these redox changes are not an essential part of the HX signaling events that activate carotid sinus chemoafferents, and the direct application of reduced L,D-DTT, reduced glutathione, or cytochrome P-450 inhibitors do not affect the expression of HX-induced chemosensory responses in isolated rat carotid bodies [9,48,58,59]. As such, the efficacy of L,D-DTT may be related to its ability to act outside the carotid bodies (e.g., the central nervous system), or perhaps as a dithiol structure, L,D-DTT exerts effects that monothiols such as glutathione and L-NACme cannot. The finding that L,D-DTT markedly decreased HH-induced increases in PIF and PEF suggests that central and/or peripheral (e.g., neuro-muscular junction) systems that drive changes in ventilatory mechanics during inspiration and expiration are sensitive to L,D-DTT. Moreover, the ability of L,D-DTT to blunt the elevations in InspD and ExpD during HH gas challenge were largely due to the processes that blunt the increases in TV. Finally, the minimal effects of the HH gas challenge and L,D-DTT on NEBI certainly suggests that L,D-DTT-sensitive (including redox) processes are not activated during HH gas challenges. 

The pattern of ventilatory responses that occurred upon return to room-air following the HH gas challenge was consistent with our previous report [16], and represents a hyper-dynamic phase of breathing that is independent of overt behaviors [10,16]. The ventilatory responses in the vehicle-injected rats included (1) sustained increases in Freq with lesser sustained increases in TV that culminated in sustained increases in MV, (2) a gradual recovery of Ti to baseline values coupled with a quicker recovery of Te, (3) a slow recovery of PIF toward baseline with a more rapid recovery of PEF, (4) a slow recovery of inspiratory drive to baseline with a more rapid recovery of expiratory drive, and (5) abrupt and substantial increases in NEBI and NEBI/Freq of 10–15 min in duration. Major findings with respect to the L,D-DTT-treated rats were that the changes in Freq, MV, Ti, Te, PIF, PEF, NEBI, and NEBI/Freq upon the return to room-air were markedly ameliorated compared to those in the vehicle-treated rats. As such, it is evident that L,D-DTT-sensitive processes, including those responsive to the potential formation of S-nitroso-D,L-DTT [95], play an essential role in the expression of the return-to-room-air ventilatory responses. The absence of increases in NEBI and NEBI/Freq upon return to room-air in the L,D-DTT-treated rats raises the intriguing possibility that novel therapeutics, which have pharmacological properties similar to L,D-DTT and S-nitroso-L,D-DTT may be of benefit to treat disorders that are associated with a substantial degree of non-eupneic breathing events, such as central/obstructive sleep apnea [66,67]. Indeed, redox-modulating therapeutics that reduce non-eupneic breathing events, such as apneas and disordered breaths, may be novel drugs to treat irregular breathing patterns, such as sleep apnea [66,67] and in subjects using opioids, which markedly destabilize breathing [19,20].

### Study Limitations

It should be noted that there are several limitations to this study. The first limitation pertains to the question as to whether the actions of L,D-DTT involves the reduced form or whether conversion to the oxidized form (Figure 1) is important to its activity/efficacy. Moreover, cell and molecular studies that provide deeper mechanistic insights as to the extracellular and intracellular targets of L,D-DTT are needed to understand the precise mechanisms of action and biochemical pathways involved in the in vivo actions of this compound. One such possibility is that L,D-DTT is nitrosylated to S-nitroso-L,D-DTT, which inhibits Kv1.1-1.6 channels via binding at internal sites of the channels [95]. The oxidized form is commercially available, and we are currently testing its activity in protocols involving HH gas challenge and return to room-air in male and female Sprague Dawley rats. An obvious limitation is that we need to test doses higher than 100 μmol/kg used in these studies for potential additional beneficial effects, and varying doses of DTT analogues that may have more potency and efficacy. Another important limitation we are currently addressing is our lack of information as to the effects of HH gas challenge and return to room-air in female Sprague Dawley rats, and whether the reduced and/or oxidized forms of L,D-DTT or analogues have biological activity in females. To better understand the mechanisms of action of L,D-DTT in our present set of findings, it is imperative that we determine how the reduced and oxidized forms of L,D-DTT modulate the effects of pure HX and pure HC gas challenges to determine whether L,D-DTT is broadly active or whether it has specific effects on HX or HC signaling pathways.

## 5. Conclusions

In summary, the present study demonstrates that many of the processes which drive the ventilatory responses that occur during a HH gas challenge and those upon return to room-air are highly sensitive to L,D-DTT. These novel findings extend our knowledge about the biochemical/pharmacological properties of L,D-DTT [60,61,62,63]. The ability of L,D-DTT to virtually eliminate the increase in non-eupneic breathing that occurs upon return to room-air is particularly provocative and raises the possibility of its use in conditions associated with unstable breathing patterns. There is now substantial evidence that L-NACme [76,77,78,79,80,81,82,83,84] and N-acetyl-L-cysteine ethyl ester [79,80,97] have important pharmacological properties. However, the present findings suggest that L-NACme is not effective at modulating the ventilatory responses to HH gas challenge or those upon return to room-air. Whether the biological efficacy of L,D-DTT involves its reduced and/or S-nitrosylated form or its oxidized form remains to be established. Moreover, an important question arises as to whether the activity of L,D-DTT depends on the L- or D- or indeed both stereoisomeric forms. This study is the first to provide compelling evidence that intravenous administration of the reduced form L,D-DTT to unrestrained, unanesthetized adult male Sprague Dawley rats has remarkable actions on the ventilatory responses that occur during HH gas challenge and upon return to room-air. The finding that the bolus injection of L,D-DTT is capable of exerting pronounced effects on the latter stages of HH gas challenge and especially the return-to-room-air responses, which are initiated 75 min after injection, is truly remarkable. In particular, the ability of L,D-DTT to virtually eliminate the increase in non-eupneic breathing that occurs upon return to room-air is a small start to determining whether L,D-DTT may be a therapeutic for the treatment of conditions associated with disordered breathing patterns, such as in patients with central and/or obstructive sleep apneas [66,67] and in subjects using opioids [19,20]. The value of this work will be brought into better perspective when the planned studies with oxidized L,D-DTT are completed.

## Figures and Tables

**Figure 1 antioxidants-13-00498-f001:**
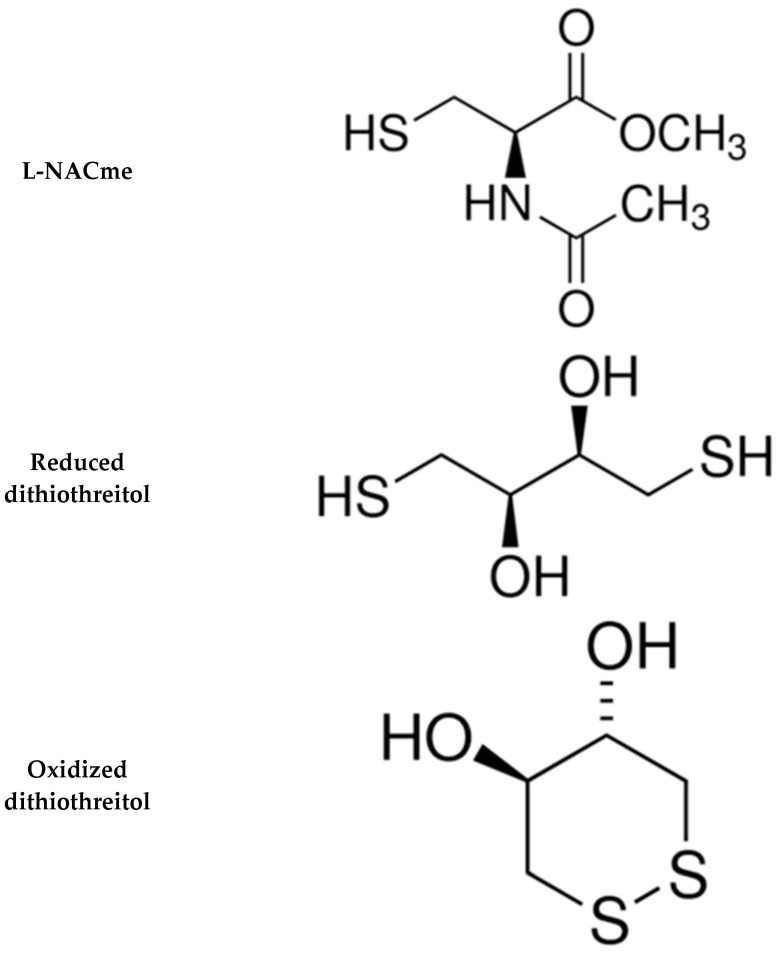
Chemical structures of N-acetyl-L-cysteine methyl ester (L-NACme) and the reduced and oxidized forms of dithiothreitol.

**Figure 2 antioxidants-13-00498-f002:**
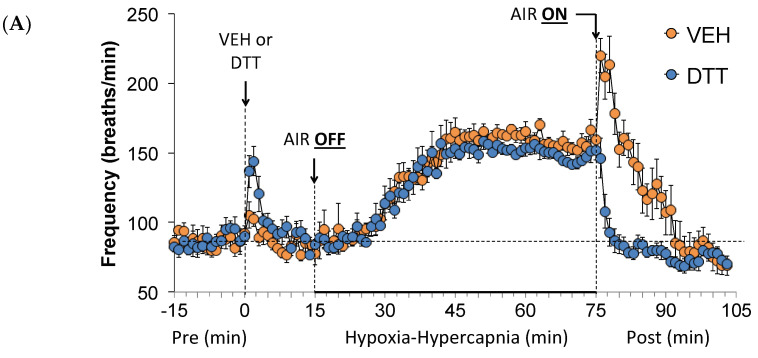
A summary of the values for frequency of breathing (**A**), tidal volume (**B**), and minute ventilation (**C**) before (Pre) and after injection of vehicle (VEH) or dithiothreitol (DTT, 100 µmol/kg, IV) and subsequent hypoxic–hypercapnic (HH) gas challenge (AIR OFF), and upon return to room air (AIR ON) in freely moving Sprague Dawley rats. The data are presented as mean ± SEM. There were six rats in each group.

**Figure 3 antioxidants-13-00498-f003:**
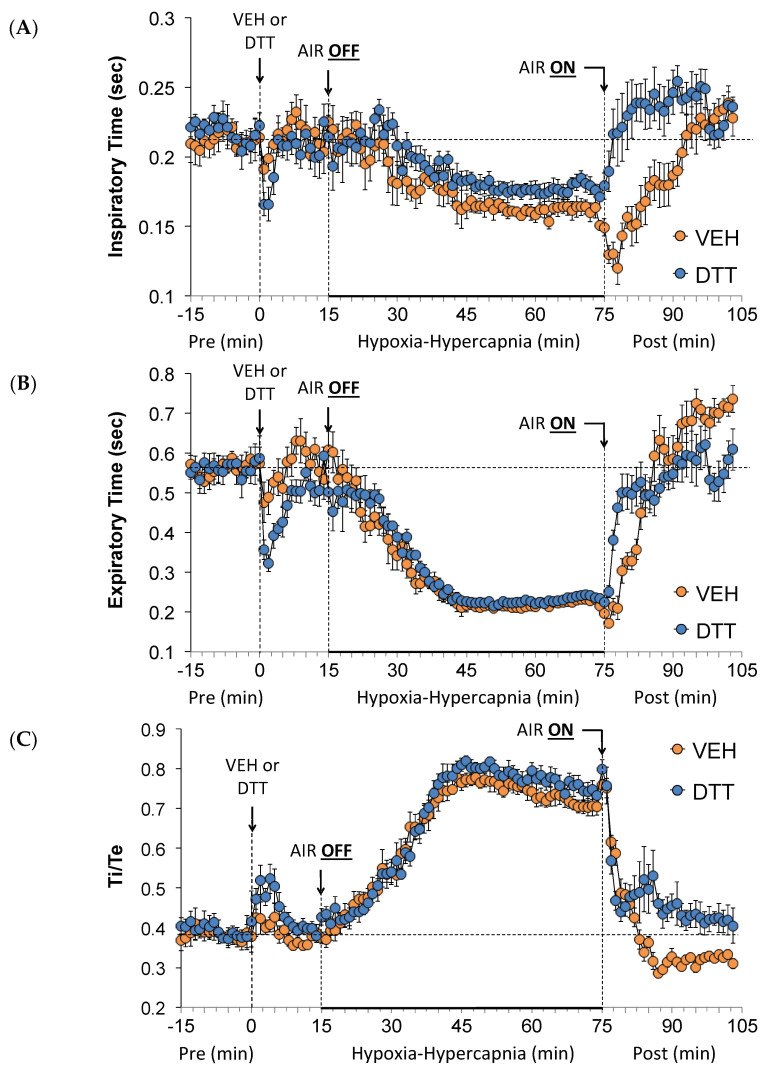
A summary of the values for inspiratory time (**A**), expiratory time (**B**), and inspiratory time (Ti)/expiratory time (Te) (**C**) before (Pre), and after injection of vehicle (VEH) or dithiothreitol (DTT, 100 µmol/kg, IV) and subsequent hypoxic–hypercapnic (HH) gas challenge (AIR OFF), and upon return to room air (AIR ON) in freely moving Sprague Dawley rats. The data are presented as mean ± SEM. There were 6 rats in each group.

**Figure 4 antioxidants-13-00498-f004:**
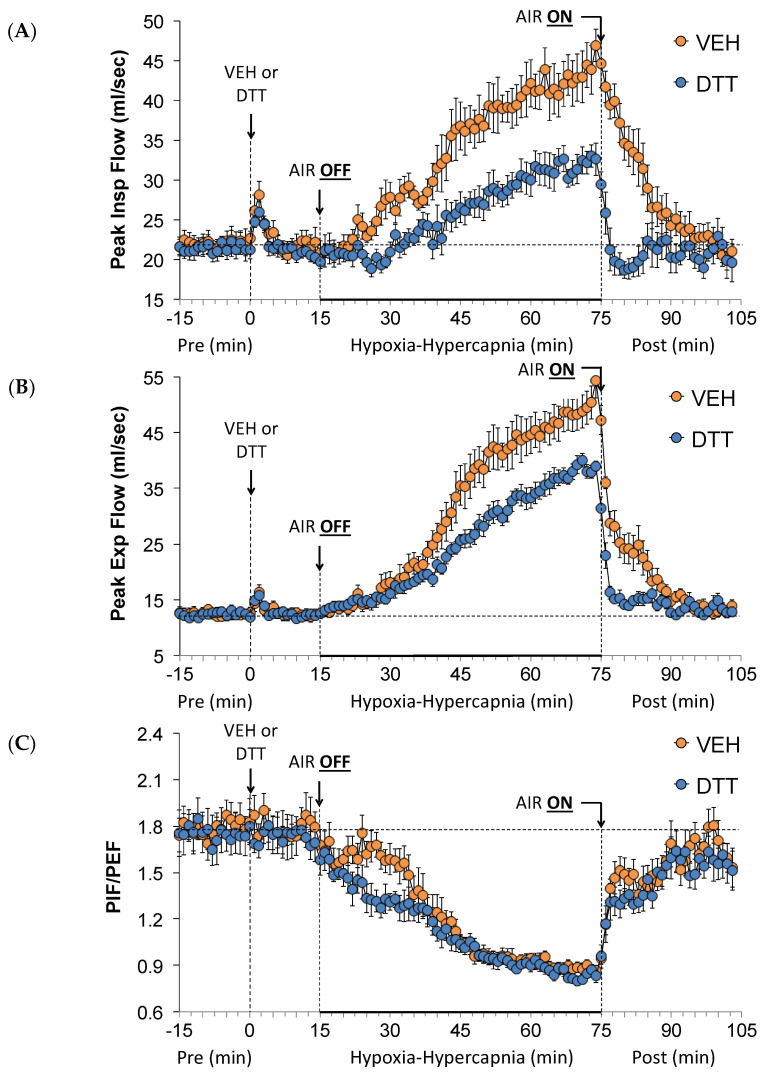
A summary of the values for peak inspiratory flow (**A**), peak expiratory flow (**B**), and peak inspiratory flow (PIF)/peak expiratory flow (PEF) (**C**) before (Pre) and after injection of vehicle (VEH) or dithiothreitol (DTT, 100 µmol/kg, IV) and subsequent hypoxic–hypercapnic (HH) gas challenge (AIR OFF), and upon return to room air (AIR ON) in freely moving Sprague Dawley rats. The data are presented as mean ± SEM. There were 6 rats in each group.

**Figure 5 antioxidants-13-00498-f005:**
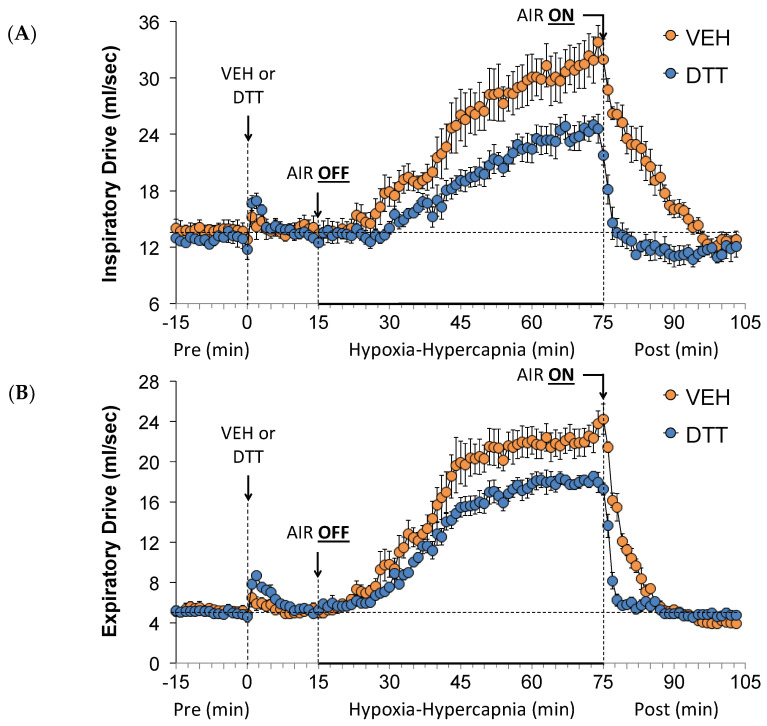
A summary of the values for inspiratory drive (**A**) and expiratory drive (**B**) before (Pre) and after the injection of vehicle (VEH) or dithiothreitol (DTT, 100 µmol/kg, IV) and the subsequent hypoxic–hypercapnic (HH) gas challenge (AIR OFF), and upon return to room air (AIR ON) in freely moving Sprague Dawley rats. The data are presented as mean ± SEM. There were 6 rats in each group.

**Figure 6 antioxidants-13-00498-f006:**
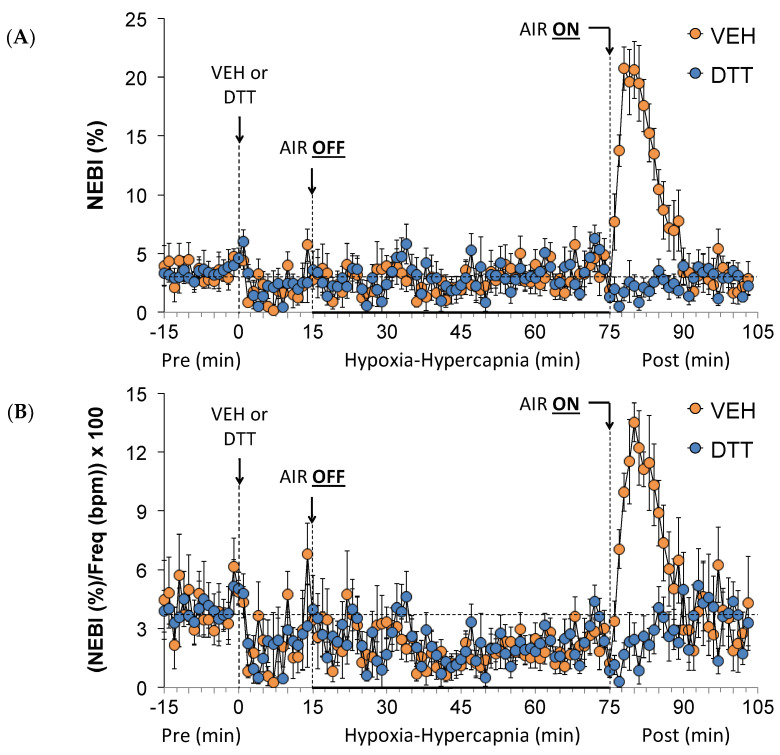
A summary of the values for non-eupneic breathing index (NEBI) (**A**) and NEBI divided by frequency of breathing (NEBI/Freq) (**B**) before (Pre) and after the injection of vehicle (VEH) or dithiothreitol (DTT, 100 µmol/kg, IV) and subsequent hypoxic–hypercapnic (HH) gas challenge (AIR OFF) and upon return to room air (AIR ON) in freely moving Sprague Dawley rats. The data are presented as mean ± SEM. There were 6 rats in each group.

**Figure 7 antioxidants-13-00498-f007:**
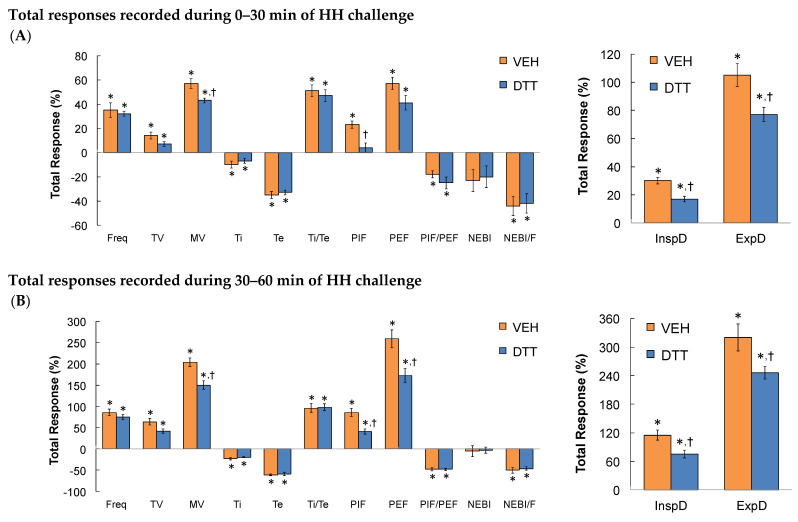
A summary of the total responses that occurred during the first 30 min of hypoxic–hypercapnic (HH) gas challenge (**A**) and the second 30 min of HH gas challenge (**B**) in rats that received an injection of vehicle (VEH) or dithiothreitol (DTT, 100 µmol/kg, IV). The data are presented as mean ± SEM. There were 6 rats in each group. * *p* < 0.05, significant change from Pre-values. ^†^
*p* < 0.05, DTT versus VEH.

**Figure 8 antioxidants-13-00498-f008:**
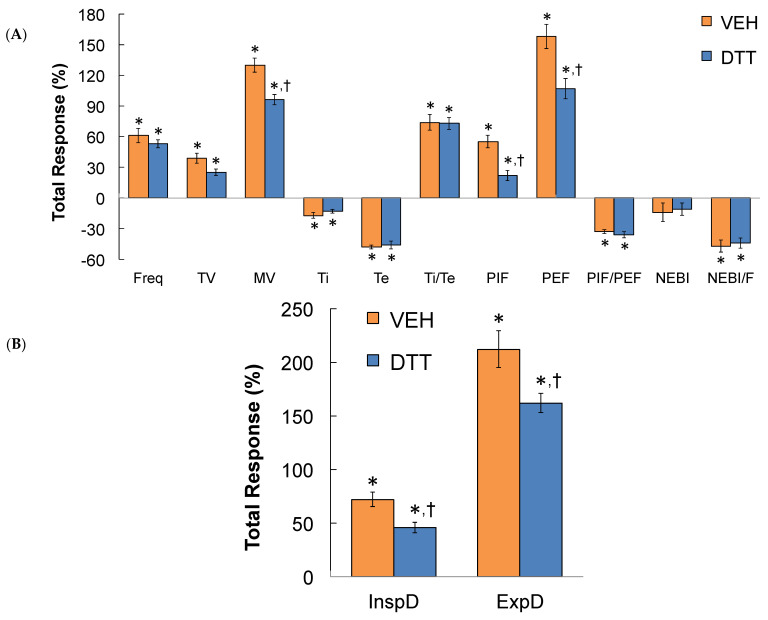
A summary of the total responses that occurred during the entire 60 min of hypoxic–hypercapnic (HH) gas challenge (**A**,**B**) in rats that received an injection of vehicle (VEH) or dithiothreitol (DTT, 100 µmol/kg, IV). Data are shown as mean ± SEM. There were 6 rats in each group. * *p* < 0.05, significant change from Pre-values. ^†^
*p* < 0.05, DTT versus VEH.

**Figure 9 antioxidants-13-00498-f009:**
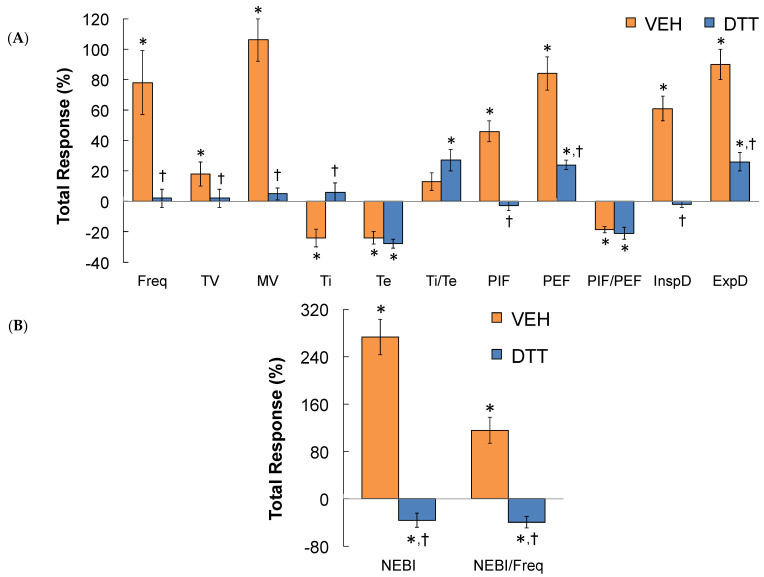
A summary of the total responses that occurred during the 30 min period following return to room-air after a 60 min hypoxic–hypercapnic (HH) gas challenge (**A**,**B**) in rats that were injected with vehicle (VEH) or dithiothreitol (DTT, 100 µmol/kg, IV). The data are presented as mean ± SEM. There were 6 rats in each group. * *p* < 0.05, significant change from Pre-values. ^†^
*p* < 0.05, DTT versus VEH.

**Figure 10 antioxidants-13-00498-f010:**
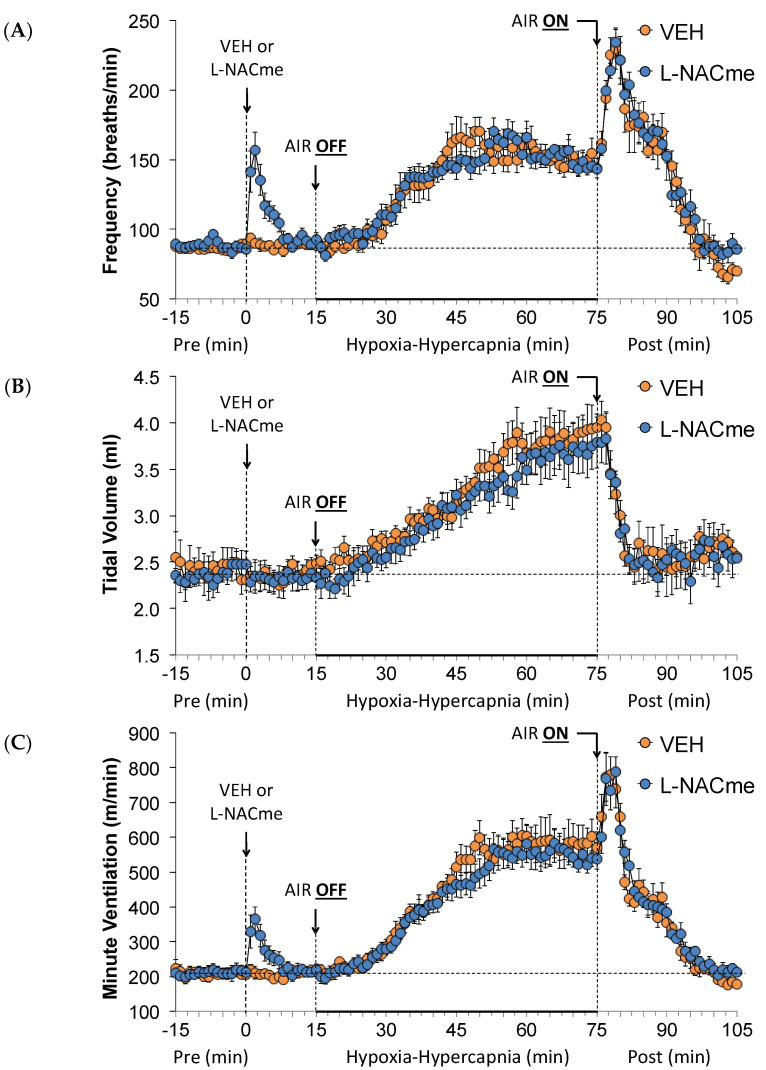
A summary of the values for frequency of breathing (**A**), tidal volume (**B**), and minute ventilation (**C**) before (Pre) and after injection of vehicle (VEH) or N-acetyl-L-cysteine methyl ester (L-NACme, 100 µmol/kg, IV) and subsequent hypoxic–hypercapnic (HH) gas challenge (AIR OFF), and upon return to room air (AIR ON) in freely moving Sprague Dawley rats. The data are presented as mean ± SEM. There were 6 rats in each group.

**Figure 11 antioxidants-13-00498-f011:**
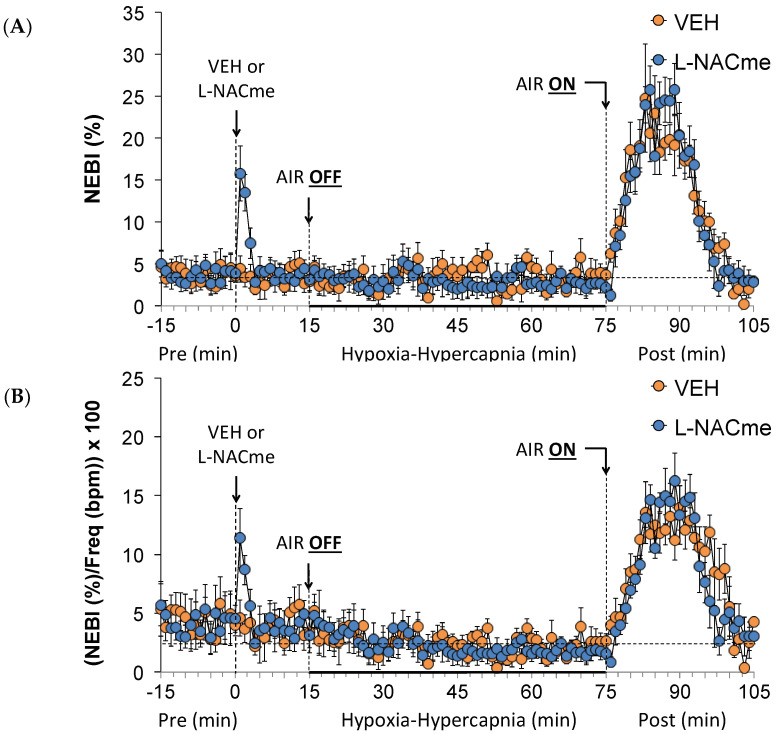
A summary of the values for non-eupneic breathing index (NEBI) (**A**) and NEBI divided by frequency of breathing (NEBI/Freq) (**B**) before (Pre) and after the injection of vehicle (VEH) or N-acetyl-L-cysteine methyl ester (L-NACme, 500 µmol/kg, IV) and subsequent hypoxic–hypercapnic (HH) gas challenge (AIR OFF), and upon return to room air (AIR ON) in freely moving Sprague Dawley rats. The data are presented as mean ± SEM. There were 6 rats in each group.

**Table 1 antioxidants-13-00498-t001:** Definition of ventilatory parameters described in this study.

Parameters	Abbreviation	Units	Definition
A. Directly recorded parameters			
Frequency of breaths	Freq	breaths/min	Rate of breathing
Inspiratory time	Ti	s	Duration of inspiration
Expiratory time	Te	s	Duration of expiration
Tidal volume	TV	mL	Volume of inspired air per breath
Minute ventilation	MV = freq × TV	mL/min	Total volume of air inspired per min
Peak inspiratory flow	PIF	mL/s	Maximum inspiratory flow
Peak expiratory flow	PEF	mL/s	Maximum expiratory flow
Non-eupneic breathing index	NEBI	%	% of non-eupneic breaths per epoch
B. Derived parameters			
Ti/Te	Ti/Te	none	Inspiratory quotient
Inspiratory drive	TV/Ti	mL/s	Central urge to inhale
Expiratory drive	TV/Te	mL/s	Central drive to exhale
PIF/PEF	PIF/PEF	none	Flow balance
NEBI/Freq	NEBI corrected for Freq	%/(breaths/min)	Expiratory ratio

**Table 2 antioxidants-13-00498-t002:** Body temperature changes during various stages of the experiment.

Phase	Time (min)	Vehicle	L,D-DTT
Pre-drug		37.4 ± 0.1	37.3 ± 0.1
Post drug	+1 min	37.3 ± 0.1	37.3 ± 0.1
	+5 min	37.4 ± 0.1	37.0 ± 0.2
	+15 min	37.3 ± 0.1	37.0 ± 0.1 *^,†^
HH gas challenge	+15 min	37.3 ± 0.1	36.9 ± 0.1 *^,†^
	+30 min	37.2 ± 0.2	36.9 ± 0.1 *
	+45 min	37.1 ± 0.1	36.9 ± 0.1 *
	+60 min	37.0 ± 0.1 *	36.9 ± 0.1 *
Post-HH gas challenge	+1 min	37.0 ± 0.2 *	36.9 ± 0.1 *
	+5 min	37.2 ± 0.2	37.1 ± 0.2
	+15 min	37.3 ± 0.1	37.3 ± 0.2
	+30 min	37.4 ± 0.2	37.3 ± 0.1

HH, hypoxia–hypercapnia. All data are presented as mean ± SEM. There were 6 rats in each group. * *p* < 0.05, change from pre-drug values. ^†^
*p* < 0.05, L,D-dithiothreitol versus vehicle.

**Table 3 antioxidants-13-00498-t003:** Ventilatory parameters at key points of the study.

Parameter	Group	Pre	Drug Max	Pre-HH	HH Max	Pre-RA	RA Max
Freq, breaths/min	VEH	87 ± 4	105 ± 9 *	82 ± 7	174 ± 8 *	160 ± 6 *	220 ± 12 *
	DTT	86 ± 6	144 ± 10 *^,†^	83 ± 4	151 ± 8 *^,†^	150 ± 6 *	146 ± 14 *^,†^
TV, mL	VEH	2.90 ± 0.20	3.07 ± 0.21	2.99 ± 0.23	5.14 ± 0.25 *	4.99 ± 0.26	3.71 ± 0.18 *
	DTT	2.78 ± 0.22	2.98 ± 0.19	2.72 ± 0.12	4.39 ± 0.14 *^,†^	4.14 ± 0.14 *	3.34 ± 0.15 *
MV, mL/min	VEH	249 ± 19	299 ± 20 *	235 ± 21	843 ± 48 *	797 ± 46 *	806 ± 21 *
	DTT	239 ± 17	394 ± 18 *	223 ± 8	645 ± 25 *^,†^	619 ± 26 *^,†^	486 ± 46 *^,†^
Ti, s	VEH	0.21 ± 0.01	0.19 ± 0.01	0.21 ± 0.01	0.15 ± 0.01 *	0.15 ± 0.01	0.12 ± 0.01 *
	DTT	0.22 ± 0.01	0.16 ± 0.01 *^,†^	0.21 ± 0.01	0.17 ± 0.01 *	0.18 ± 0.01	0.17 ± 0.01 *^,†^
Te, s	VEH	0.56 ± 0.08	0.48 ± 0.05	0.57 ± 0.05	0.20 ± 0.02 *	0.21 ± 0.01 *	0.17 ± 0.01 *
	DTT	0.55 ± 0.07	0.32 ± 0.02 *^,†^	0.53 ± 0.06	0.24 ± 0.02 *	0.23 ± 0.01 *	0.25 ± 0.02 *^,†^
Ti/Te,	VEH	0.38 ± 0.02	0.43 ± 0.04 *	0.37 ± 0.02	0.78 ± 0.02 *	0.76 ± 0.02 *	0.29 ± 0.01 *
	DTT	0.39 ± 0.012	0.52 ± 0.04 *	0.43 ± 0.02	0.82 ± 0.02 *	0.80 ± 0.02 *	0.41 ± 0.04 *^,†^
PIF, mLs/s	VEH	22.0 ± 1.2	28.1 ± 1.7 *	21.6 ± 1.6	46.9 ± 2.0 *	45.1 ± 2.7 *	41.7 ± 2.0 *
	DTT	21.4 ± 1.2	26.0 ± 1.3 *	20.2 ± 0.8	32.6 ± 2.1 *^,†^	31.7 ± 1.7 *^,†^	29.4 ± 1.6 *^,†^
PEF, mLs/s	VEH	12.5 ± 0.9	16.3 ± 1.3 *	12.3 ± 0.9	54.4 ± 3.1 *	50.7 ± 2.9 *	38.0 ± 2.1 *
	DTT	12.4 ± 0.6	15.7 ± 1.2 *	12.4 ± 0.7	40.0 ± 2.3 *^,†^	36.1 ± 1.4 *^,†^	29.4 ± 2.1 *^,†^
PIF/PEF	VEH	1.79 ± 0.08	1.90 ± 0.11	1.68 ± 0.10	0.87 ± 0.04 *	0.95 ± 0.04 *	38.0 ± 2.1 *
	DTT	1.75 ± 0.11	1.80 ± 0.11	1.58 ± 0.10	0.80 ± 0.03 *	0.96 ± 0.08 *	29.4 ± 2.1 *

All data are presented as mean ± SEM. There were 12 rats in each group. VEH, vehicle; DTT, dithiothreitol; Freq, frequency of breathing; TV, tidal volume; MV, minute volume; Ti, inspiratory time; Te, expiratory time; PIF, peak inspiratory flow; PEF, peak expiratory flow. Note that there were no between-group differences in any parameter prior to administration of VEH or DTT (*p* > 0.05, for all comparisons). * *p* < 0.05, significant change from Pre-values. ^†^
*p* < 0.05, DTT versus VEH.

**Table 4 antioxidants-13-00498-t004:** Ventilatory parameters at key points of the study.

Parameter	Group	Pre	Drug Max	Pre-HH	HH Max	Pre-RA	RA Max
TV/Ti, mL/s	VEH	13.7 ± 0.9	15.2 ± 1.1	13.8 ± 1.1	33.8 ± 1.8 *	32.6 ± 2.1 *	28.7 ± 2.2 *
	DTT	12.8 ± 0.6	16.9 ± 0.8 *	12.9 ± 0.5	25.0 ± 0.9 *^,†^	24.0 ± 1.2 *^,†^	18.1 ± 1.5 *^,†^
TV/Te, mL/s	VEH	5.3 ± 0.4	6.5 ± 0.6	5.0 ± 0.4	24.2 ± 1.5 *	24.2 ± 1.5	3.9 ± 0.3 *
	DTT	5.0 ± 0.2	8.7 ± 0.4 *^,†^	5.3 ± 0.3	25.0 ± 0.9 *	17.3 ± 0.6 *^,†^	4.5 ± 0.2
NEBI, %	VEH	3.46 ± 0.23	0.17 ± 0.17 *	2.75 ± 0.77	0.92 ± 0.49 *	1.92 ± 1.1	20.72 ± 1.82 *
	DTT	3.39 ± 0.06	0.40 ± 0.19 *	3.58 ± 1.60	18.4 ± 0.6 *	1.83 ± 0.64	3.52 ± 0.98 ^†^
NEBI/Freq, %/bpm	VEH	4.05 ± 0.35	0.23 ± 0.23 *	3.43 ± 0.65	0.70 ± 0.37 *	1.21 ± 0.70 *	13.51 ± 0.92 *
	DTT	3.99 ± 0.13	0.44 ± 0.20 *	3.95 ± 0.76	0.68 ± 0.63 *	0.85 ± 0.32 *	5.18 ± 0.91 ^†^

VEH, vehicle; DTT, dithiothreitol; TV, tidal volume; Ti, inspiratory time; Te, expiratory time; NEBI, non-eupneic breathing index. Freq, frequency of breathing. All data are presented as mean ± SEM. There were 6 rats in each group. Note that there were no between-group differences in any parameter prior to administration of VEH or DTT (*p* > 0.05, for all comparisons). * *p* < 0.05, significant change from Pre-values. ^†^
*p* < 0.05, DTT versus VEH.

**Table 5 antioxidants-13-00498-t005:** Time to return to baseline levels following return to room-air.

	Time to Return to Pre-HH Levels (min)	
Parameter	Vehicle	L,D-Dithiothreitol
Frequency, breaths/min	16.3 ± 1.9	4.1 ± 1.0 *
Tidal Volume, mL	15.2 ± 3.2	4.5 ± 0.9 *
Minute Volume, mL/min	18.3 ± 1.4	5.6 ± 0.8 *
Inspiratory Time, s	17.4 ± 2.4	3.4 ± 0.8 *
Expiratory Time, s	4.0 ± 1.1	9.2 ± 1.3 *
Inspiratory Time/Expiratory Time	7.6 ± 0.8	13.3 ± 1.1 *
Tidal Volume/Inspiratory Time, mL/s	17.8 ± 1.5	1.9 ± 0.6 *
Tidal Volume/Expiratory Time, mL/s	7.4 ± 0.6	16.2 ± 1.9 *
Peak Inspiratory Flow, mL/s	18.6 ± 2.5	2.1 ± 0.4 *
Peak Expiratory Flow, mL/s	14.5 ± 2.1	4.7 ± 0.9 *
Non-Eupneic Breathing Index (NEBI), %	14.3 ± 1.7	N/D *
NEBI, %/Freq, breaths/min	14.7 ± 1.9	N/D *

HH, hypoxia–hypercapnia, N/D, not determinable. All data are presented as mean ± SEM. There were 12 rats in each group. * *p* < 0.05, dithiothreitol versus vehicle.

## Data Availability

The corresponding author will provide the datasets generated from this study upon email request to sjl78@case.edu.

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
