# Peer review of "The Reducing Agent Dithiothreitol Modulates the Ventilatory Responses That Occur in Freely Moving Rats during and following a Hypoxic–Hypercapnic Challenge"

_antioxidants, 2024, doi:10.3390/antiox13040498_

Round 1
Reviewer 1 Report
The manuscript by Paulina M. Getsy et al. investigates the effects of dithiothreitol (DTT) and N-acetyl-L-cysteine methyl ester (L-NACme) on ventilatory responses in rats during and after a hypoxic-hypercapnic (HH) challenge. The study is comprehensive, with detailed methodologies, significant findings, and a thorough discussion of the implications of these findings, while there are a few concerns should be noted:
1) Please clarify if there is any clinical relevance or significance in the introduction or discussion.
2) The authors labeled 6 rats for each experiment. Are they from a batch of experiments or pooled from different batches? If the data is from one representative experiment, how many times has it been repeated? While the study uses vehicle controls, including additional controls, such as animals treated with other known antioxidants or reducing agents, could help contextualize the specificity and potency of DTT's effects.
3)While the study uses vehicle controls, including additional controls, such as animals treated with other known antioxidants or reducing agents, could help contextualize the specificity and potency of DTT's effects.
4)Although the manuscript discusses several caveats and limitations throughout, consolidating these into a dedicated Limitation Section could help readers better appreciate the study's scope and the caution needed in interpreting the results.
NA
Author Response
Reviewer 1
The manuscript by Paulina M. Getsy et al. investigates the effects of dithiothreitol (DTT) and N-acetyl-L-cysteine methyl ester (L-NACme) on ventilatory responses in rats during and after a hypoxic-hypercapnic (HH) challenge. The study is comprehensive, with detailed methodologies, significant findings, and a thorough discussion of the implications of these findings, while there are a few concerns should be noted:
Reply
We appreciate these comments and have tried hard to respond to your individual comments.
Point 1
1) Please clarify if there is any clinical relevance or significance in the introduction or discussion.
Reply
We have revised the last sentence in the Conclusion section. The text now reads, “The absence of increases in NEBI and NEBI/Freq upon return to room-air in L,D-DTT-treated rats raises the intriguing possibility that novel therapeutics which have pharmacological properties similar to L,D-DTT and S-nitroso-L,D-DTT may be of benefit to treat disorders that are associated with a substantial degree of non-eupneic breathing events such as central/obstructive sleep apnea [66,67]. Indeed, redox-modulating therapeutics that reduce non-eupneic breathing events such as apneas and disordered breaths may be novel drugs to treat irregular breathing patterns such as sleep apnea [see 66,67] and in subjects using opioids, which markedly destabilize breathing [see 19,20].”
Point 2
2) The authors labeled 6 rats for each experiment. Are they from a batch of experiments or pooled from different batches? If the data is from one representative experiment, how many times has it been repeated?
Reply
This is an excellent point. The DTT studies were performed over a 3 month period using two different batches of rats. As such 3 rats were given vehicle and 3 rats were given DTT from each batch. The consistency of the data makes us believe that the findings are robust. As part of on-going studies comparing DTT with analogues of DTT we have replicated the findings with DTT in male (n=6) and female (n=6) rats (manuscript in preparation). We added the following sentence to the Methods section, “Note that two separate baches of 3 rats per group were used in each of the above studies to help ensure that the data was reproducible.”
Point 3
3) While the study uses vehicle controls, including additional controls, such as animals treated with other known antioxidants or reducing agents, could help contextualize the specificity and potency of DTT's effects.
Reply
Yes we agree completely. The data in Figure 10 and Figure 11 showing that a large dose of the potent (mono-thiol) reducing agent N-acetyl-L-cysteine ethyl ester (L-NACme) had negligible effects on the hypoxic-hypercapnic or return to room-air responses speaks to the uniqueness of the di-thiol, DTT.
Point 4
4) Although the manuscript discusses several caveats and limitations throughout, consolidating these into a dedicated Limitation Section could help readers better appreciate the study's scope and the caution needed in interpreting the results.
Reply
We have a section labeled “Study Limitation” which we believe addresses this excellent point.
Reviewer 2 Report
The current article explores the role of DTT in modulating ventilatory responses in rats during and after hypoxic/hypercapnic challenges. It provides valuable insights into the redox mecanisms underlying ventilatory control and offers a fondation for further research in this area.
Here's several major concerns that need to be addressed:
-
-The methodology section should be expanded to include more detailed explanations of the experimental setup and procedures to ensure reproducibility.
-
-
-While the study suggests a role for DTT in ventilatory responses, deeper mechanistic investigations are necessary to understand the precise biochemical pathways involved. Please address this point in discussion.
-
- data analysis section might not be so easily understandable: further clarification and justification of the statistical methods used could enhance the credibility of the findings.
-
-The discussion should be expanded to better contextualize the findings within the broader field of respiratory physiology and potential clinical implications.
Author Response
Reviewer 2
The current article explores the role of DTT in modulating ventilatory responses in rats during and after hypoxic/hypercapnic challenges. It provides valuable insights into the redox mechanisms underlying ventilatory control and offers a foundation for further research in this area.
Here's several major concerns that need to be addressed:
Point 1
-The methodology section should be expanded to include more detailed explanations of the experimental setup and procedures to ensure reproducibility.
Reply
Excellent point. We added several sentences to better explain the procedures. Such sentences were “The rats explored the chambers for 5-10 min before settling in a resting position up against the chamber wall.” and “Note that two separate baches of 3 rats per group were used in each of the above studies to help ensure that the data was reproducible.”
Point 2
-While the study suggests a role for DTT in ventilatory responses, deeper mechanistic investigations are necessary to understand the precise biochemical pathways involved. Please address this point in discussion.
Reply
Yes, excellent point. We have added the following sentences to the Study Limitations section of the Discussion, “Moreover, cell and molecular studies that provide deeper mechanistic insights as to the extracellular and intracellular targets of L,D-DTT are needed to understand the precise mechanisms of action and biochemical pathways involved in the in vivo actions of this compound. One such possibility is that L,D-DTT is nitrosylated to S-nitroso-L,D-DTT which inhibits Kv1.1-1.6-channels via binding at internal sites of the channels [95].”
Point 3
- data analysis section might not be so easily understandable: further clarification and justification of the statistical methods used could enhance the credibility of the findings.
Reply
We have provided detailed explanations of the data analyses in recent publications [27-30]. In response to a request from the editor we have abbreviated and hopefully better explained the analyses. The text now reads, “All of the data collected in these studies are presented as mean ± SEM. The data from each set of experiments were statistical analyzed by one-way or two-way ANOVA followed by Bonferroni corrections for multiple comparisons between means using the error mean square terms from each individual ANOVA analysis [91-93] as described in detail previously [27-30]. A P < 0.05 value provided the initial level of statistical significance that was modified according to the number of comparisons between means as described in detail by Wallenstein et al [91]. The statistical analyses were done using GraphPad Prism software (GraphPad Software, Inc., La Jolla, CA).
Point 4
-The discussion should be expanded to better contextualize the findings within the broader field of respiratory physiology and potential clinical implications.
Reply
Yes, excellent comment. We have modified the last sentence of the Conclusion section to now read, “The absence of increases in NEBI and NEBI/Freq upon return to room-air in L,D-DTT-treated rats raises the intriguing possibility that novel therapeutics which have pharmacological properties similar to L,D-DTT and S-nitroso-L,D-DTT may be of benefit to treat disorders that are associated with a substantial degree of non-eupneic breathing events such as central/obstructive sleep apnea [66,67]. Indeed, redox-modulating therapeutics that reduce non-eupneic breathing events such as apneas and disordered breaths may be novel drugs to treat irregular breathing patterns such as sleep apnea [see 66,67] and in subjects using opioids, which markedly destabilize breathing [see 19,20].”
Round 2
Reviewer 2 Report
This is a second review.
Authors replied to my comments in a satisfactorily way. Therefore, IMHO, this article can now be accepted for publication.